# Graphlet eigencentralities capture novel central roles of genes in pathways

**Sam F. L. Windels**[1,2], **Noël Malod-Dognin**[1,2], **Nataša Pržulj**[1,2,3]*

**1** Department of Computer Science, University College London, London, United Kingdom, **2** Barcelona Supercomputing Center (BSC), Barcelona, Spain, **3** ICREA, Barcelona, Spain

* natasa@bsc.es

## Abstract

### Motivation

Graphlet adjacency extends regular node adjacency in a network by considering a pair of nodes being adjacent if they participate in a given graphlet (small, connected, induced subgraph). Graphlet adjacencies captured by different graphlets were shown to contain complementary biological functions and cancer mechanisms. To further investigate the relationships between the topological features of genes participating in molecular networks, as captured by graphlet adjacencies, and their biological functions, we build more descriptive pathway-based approaches.

### Contribution

We introduce a new graphlet-based definition of eigencentrality of genes in a pathway, graphlet eigencentrality, to identify pathways and cancer mechanisms described by a given graphlet adjacency. We compute the centrality of genes in a pathway either from the local perspective of the pathway or from the global perspective of the entire network.

### Results

We show that in molecular networks of human and yeast, different local graphlet adjacencies describe different pathways (i.e., all the genes that are functionally important in a pathway are also considered topologically important by their local graphlet eigencentrality). Pathways described by the same graphlet adjacency are functionally similar, suggesting that each graphlet adjacency captures different pathway topology and function relationships. Additionally, we show that different graphlet eigencentralities describe different cancer driver genes that play central roles in pathways, or in the crosstalk between them (i.e. we can predict cancer driver genes participating in a pathway by their local or global graphlet eigencentrality). This result suggests that by considering different graphlet eigencentralities, we can capture different functional roles of genes in and between pathways.

**Data Availability Statement:** Data availability: https://gitlab.bsc.es/swindels/graphlet_ eigencentrality.

**Funding:** This study received support from the following sources: The European Research Council

(ERC) Consolidator Grant 770827 (awarded to NP); The Spanish State Research Agency AEI 10.13039/ 501100011033 grant number PID2019-105500GB-I00 (awarded to NP); and University College London Computer Science (awarded to SW). The funders had no role in study design, data collection and analysis, decision to publish, or preparation of the manuscript.

**Competing interests:** The authors have declared that no competing interests exist.

## Introduction

Biology is flooded with large scale "omic" data. Genomic, proteomic, interatomic and other data are typically modeled as *networks* (also called *graphs*). In molecular networks, nodes usually represent genes or proteins and edges represent interactions or relationships between them, such as physical bonds between the proteins (PPI), genetic interactions (GI), or co-expressions (COEX) [1, 2].

To infer new information about the genes in these networks (or equivalently, proteins), such as their participation in pathways, their biological function, or their roles in diseases, three main categories of approaches exist. *Guilt by association* approaches infer annotations of a given node (gene, or protein) based on the annotations of the nodes adjacent to it. Implicitly, these approaches are based on one of the core ideas in network biology, which considers biological function to arise from groups of genes forming functional modules [3]. In practice, many network-based learning approaches are based on a slightly more strict version of this assumption called *homophily*, which assumes that these functional modules are densely connected [4]. For instance, spectral clustering uncovers functional groups of genes by cutting the full network into densely connected subnetworks [5]. This homophily assumption is widely present in network-based learning and is not always as explicit. For instance, Hierarchical Hot-Net applies heat-diffusion of somatic mutation scores on a molecular network, which quantifies for each gene how likely it is to be somatically mutated in cancer. After diffusion, subnetworks that have maintained a lot of 'heat' are predicted as being cancer-related [6]. The nature of heat-diffusion makes it more likely that heat is retained within densely connected areas of the molecular network, thus implicitly assuming homophily.

*Centrality* based approaches infer gene annotations based on their importance in the network. Centrality measures quantify the importance of a node either based its connectivity, or based on its frequency of occurrence on shortest paths. Ever since [7] showed that perturbing highly connected nodes in PPI networks has a higher probability of impacting cell viability, these methods have become a major tool for discovering gene functions and uncovering disease-related genes (e.g., see [8–10]). For instance, 'local radiality' measures how frequently a node is found on the shortest path between genes that have their expression perturbed by a given drug and is used for drug-target prioritization [11].

Approaches based on *graphlets*, small, connected, induced subgraphs, infer new knowledge about a given node based on the annotations of nodes with a similar wiring pattern, independent of them being in the same network neighbourhood [12]. Informally, local wiring patterns are quantified by counting how often a node touches different graphlets. For instance, graphlet based methods have been applied to predict protein function [13] and identify age-related genes [14] based on their interaction patterns in PPI networks.

To combine graphlet-based topological information and network neighbourhood information, we recently introduced graphlet adjacencies, which consider a pair of nodes to be 'adjacent' if they simultaneously touch a given graphlet [15]. We have shown that graphlet adjacency for different underlying graphlets captures complementary biological functions, by performing network clustering followed by cluster enrichment analysis. Additionally, diffusing pan-cancer gene mutation scores over the human PPI network based on different types of graphlet adjacencies showed that graphlet adjacencies captures complementary disease mechanisms.

Where detailed insight is needed, network-biology resorts to studying *pathways*: functional subnetworks within the cell that once activated lead to a certain product, or a change within the cell. Pathway-based approaches are frequently used to study cancer genes, as only a few genes are frequently mutated and studying them as part of pathways provides insight into the

underlying processes. This, in turn, helps to generate testable hypotheses, identifying drug targets and tumour subtypes [16]. Three major pathway-based approaches exist. *Gene set enrichment analysis* considers pathways as gene-sets (ignoring information about interactions between the genes) and identifies pathways enriched in mutated, or differentially expressed genes. *Network-modeling-based* approaches study pathways taking topology into account. For instance, PathOlogist measures the consistency between gene expression data and pathway topology, enabling the identification of diseased pathways [17]. Both types of approaches assume pathways to be a part of prior knowledge. The function of a pathway, the genes that take part in it and the interactions between them, are assumed to be known in advance. Curated pathways can be found in the Reactome database [18]. Being based on curated biological pathways, both types of approaches lead to highly interpretable results. However, as pathway knowledge is incomplete, *de novo network construction* methods aim at uncovering and studying subnetworks significantly perturbed in disease. For instance, given a large biological interaction network, KeyPathwayMiner extracts connected subnetworks enriched in differentially expressed genes and interprets them as functional modules or *de novo* pathways.

## Problem

Although we have shown that graphlet adjacencies capture complementary biological function and cancer mechanisms, we have not provided insight into the underlying biology [15]. In particular, we concluded that different graphlet adjacencies capture complementary biological functions by showing that graphlet adjacency based network clustering leads to functionally differently enriched clusters depending on the underlying graphlet. However, we have not investigated why some functional annotations are better captured than others by each graphlet adjacency. Similarly, we concluded that different graphlet adjacencies capture complementary disease mechanisms by diffusing pan-cancer gene mutation scores over the human PPI network and finding different cancer gene predictions based on the underlying graphlet; again we have not investigated why this is the case.

## Contribution

To further investigate the relationships between the topological features of genes in molecular networks in human and yeast, as captured by graphlet adjacencies, and the biological functions of the genes, we build more descriptive pathway-based approaches. We extend eigencentrality to graphlet eigencentrality, to study the importance (centrality) of genes in pathways; either from the local pathway perspective or the global perspective of the entire network.

First, we identify the pathways that are described by each graphlet adjacency, i.e. all genes participating in a pathway are also captured as topologically important by the graphlet adjacency. To do so, we use our graphlet eigencentralities to predict which genes belong to a given pathway, considering the pathways for which we achieve the highest prediction accuracies as being described by that graphlet adjacency. We find that local pathway-based graphlet eigencentralities well predict which genes participate in a given pathway, outperforming state-of-the-art predictor GeneMANIA (validating our approach) and our global approach. To explain this result, we show that pathways, even when functionally unrelated, show large amounts of overlap. As our local approach considers each pathway as an individual entity disentangled from the full network, it is able to best capture the topological essence of a pathway. Further, we show that the pathways that are described by a given graphlet adjacency, are functionally similar, implying that each graphlet adjacency uncovers different pathway topology and function relationships. We illustrate this relationship in the 'Receptor Mediated Mitophagy'

pathway, where we show how, unlike regular adjacency, graphlet adjacencies capture the relevance of all genes in the pathway.

Second, by considering different graphlet adjacencies, from the local and global perspective, we uncover complementary sets of cancer driver genes (known to be drivers in at least one type of cancer) that are described by playing central roles in pathways and the crosstalk between them. This suggests that by considering different graphlet eigencentralities, we can capture different functional roles of genes in and between pathways. We illustrate this in the 'Formation of Senescence-Associated Heterochromatin Foci'-pathway, where we show how, unlike regular adjacency, graphlet adjacencies capture the central roles of cancer driver genes TP53 and RB1.

## Materials and methods

Network centrality measures quantify the importance of a node in a network. We consider the formal definition of eigencentrality (Section: Eigencentrality) and extend this definition to graphlet eigencentrality (Section: Graphlet eigencentrality). Next, we explain how we use graphlet eigencentralities to measure the centrality of a gene in a pathway, or its *pathway centrality*. We can measure pathway centrality from the pathway perspective (the centrality of the genes is computed on the genes known to participate in the pathway, Section: Local pathway centrality), or from the global network perspective (the centrality of the genes is computed on the full network before inducing the set of nodes corresponding to genes participating in the pathway, Section: Global pathway centrality). Finally, we explain how we use pathway centrality to predict which genes participate in a given pathway (Section: Predicting pathway participation).

### Eigencentrality

Eigencentrality considers the nodes that are highly connected to other highly connected nodes in the network to be the most important nodes [19].

Formally, for a given unweighted and undirected network $H = (V, E)$, where $V$ is the set of nodes in network $H$ and $E$ is the set of interactions between the nodes, the centrality of a node $u \in V$, $c_u$, is defined as the average of the centralities of its $n$ neighbours:

$$c_u = \frac{1}{\lambda} \sum_{v=1}^{n} c_v A_{uv}, \tag{1}$$

where $\lambda$ is a constant and $A$ is the adjacency matrix of the network. To be able to solve this equation, we write it in matrix form:

$$A\mathbf{c} = \lambda \mathbf{c}, \tag{2}$$

where $\mathbf{c}$ is the vector of centralities, $\mathbf{c} = (c_1, c_2, \ldots)$. From this, it is clear that $\mathbf{c}$ is an eigenvector of $A$ and $\lambda$ is an eigenvalue for which a non-zero eigenvector solution exists; hence the name 'eigencentrality'. Furthermore, as the entries of $\mathbf{c}$ are required to be non-negative for an interpretation as a centrality measure, it can be shown that $\mathbf{c}$ is the eigenvector corresponding to the largest eigenvalue of $A$.

Many variations of eigencentrality exist. For instance, the *Katz centrality* generalises the eigencentrality to directed networks [20]. The *contribution centrality* extends the eigencentrality by amplifying a node's centrality if it serves as a hub node connecting densely connected parts of the network [21].

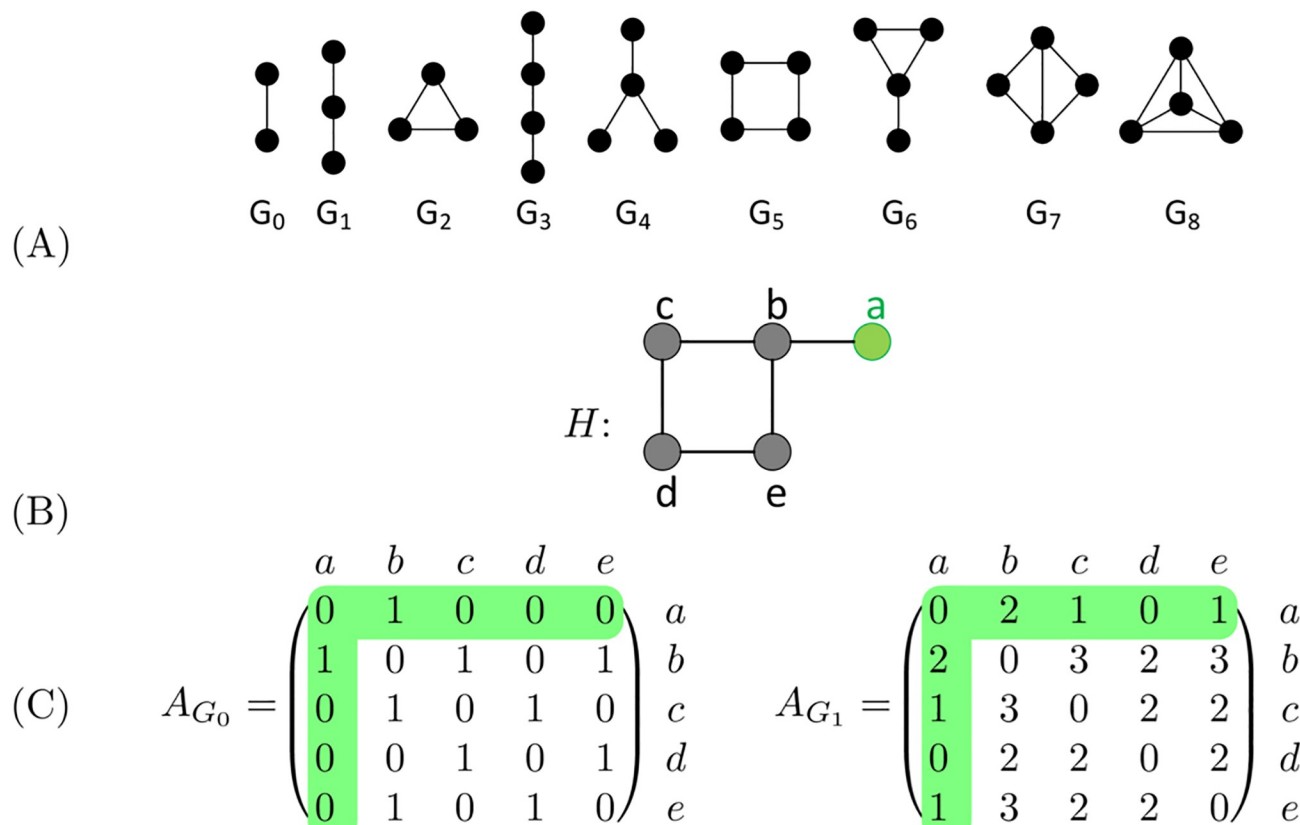

**Fig 1. An illustration of graphlets and graphlet adjacencies.** Node $a$ and its graphlet adjacencies are coloured in green throughout. **A**: All graphlets with up to 4 nodes, labelled $G_0$ to $G_8$. **B**: Example network $H$. **C**: The graphlet adjacency matrices $A_{G_0}$ and $A_{G_1}$ for graphlets $G_0$ and $G_1$ of the example network $H$, shown in panel B. The off-diagonal elements correspond to the number of times two nodes touch a given graphlet together. $A_{G_0}(a, b) = 1$, as $a$ and $b$ form $G_0$ once. $A_{G_1}(a, b) = 2$, as $a$ and $b$ form $G_1$ twice, via paths $a$-$b$-$c$ and $a$-$b$-$e$. This figure is adapted from Fig 1 in [15].

### Graphlet eigencentrality

*Graphlets* are small, connected, non-isomorphic, induced subgraphs of a large network [12]. All graphlets up to four nodes are depicted in Fig 1A. Two nodes $u$ and $v$ of $H$ are considered *graphlet adjacent* with respect to a given graphlet, $G_i$, if they simultaneously touch $G_i$ [15]. In the example network presented in Fig 1B, we find that nodes $a$ and $b$ are graphlet adjacent with respect to graphlet $G_1$ twice, as $G_1$ can be induced on the example network twice including both nodes $a$ and $b$: along paths $a$-$b$-$c$ and $a$-$b$-$e$. Given this extended definition of adjacency, the graphlet based adjacency matrix is defined as:

$$A_{G_i}(u, v) = \begin{cases} c_{uv}^{G_i}/\theta_{G_i} & \text{if } u \neq v \\ 0 & \text{otherwise,} \end{cases} \tag{3}$$

where $c_{uv}^{G_i}$ is equal to the number of times the nodes $u$ and $v$ simultaneously touch graphlet $G_i$ and $\theta_{G_i}$ is a scaling constant equal to the number of nodes in graphlet $G_i$ minus 1. Note that graphlet adjacency matrix $A_{G_0}$, is equivalent to the standard adjacency matrix. We illustrate $A_{G_0}$ and $A_{G_1}$ in Fig 1C.

Analogously, the *graphlet degree* generalizes the node degree as the number of times node $u$ touches graphlet $G_i$. The *Graphlet Degree matrix* for graphlet $G_i$, $D_{G_i}$, is defined as:

$$D_{G_i}(u, v) = \begin{cases} d_u^{G_i} & \text{if } u = v \\ 0 & \text{otherwise,} \end{cases} \tag{4}$$

where $d_u^{G_i}$ is the number of times node $u$ touches graphlet $G_i$.

The symmetrically normalised graphlet adjacency matrix is defined as:

$$\widetilde{A_{G_i}} = D_{G_i}^{-1/2} A_{G_i} D_{G_i}^{-1/2} \tag{5}$$

Intuitively, the symmetric normalisation rescales the weight of a given edge relative to its importance to both nodes involved [22].

We generalize normalised eigencentrality to graphlet eigencentrality by replacing A with the normalised graphlet adjacency matrix, $\widetilde{A_{G_i}}$, in Eq 2:

$$\widetilde{A_{G_i}} \mathbf{c}_{G_i} = \lambda_{G_i} \mathbf{c}_{G_i}, \tag{6}$$

## Pathway centrality

We aim to measure the centrality of the set of genes that participate in a given pathway. We can do this from the pathway perspective, which we will refer to as 'Local pathway centrality', or from the perspective of the entire network, which we will refer to as 'Global pathway centrality'.

**Local pathway centrality.**   We take the submatrix of the adjacency matrix of the full network corresponding to the $m$ genes participating in the pathway, to create the $m \times m$ dimensional local adjacency matrix $P$. Then, for a given underlying graphlet $G_i$, we compute the corresponding graphlet adjacency matrix, $P_{G_i}$, and compute the normalised graphlet eigencentrality applying Eq 6.

**Global pathway centrality.**   For a given underlying graphlet $G_i$, we compute the global graphlet eigencentrality vector, $\mathbf{c}_{G_i}$, on the normalised graphlet adjacency matrix, $\widetilde{A_{G_i}}$, applying Eq 6. Then, we take the subvector of the global eigenvector corresponding the $m$ to genes participating in the pathway, $\mathbf{c}_{G_i}$, to determine their pathway centrality.

## Predicting pathway participation

Pathways are functional subgraphs in which a group of genes work together to perform a given biological function. For a pathway to perform its function, each gene is essential. We consider a pathway to be described by a given graphlet adjacency if the topology captured by it correctly recognises that all genes in the pathway are important to performing its function. To evaluate which pathways are described by a given graphlet adjacency, we use our local and global pathway centrality eigencentralities to predict which genes belong to a given pathway, as described below. We consider the pathways for which we achieve the highest prediction accuracies as being described by that graphlet adjacency; as for those pathways we can best distinguish between the genes are relevant w.r.t. the pathway and those are not. To show our approach captures biological signal, we compare our prediction accuracy to that of gene annotation predictor GeneMANIA, [23].

Given a molecular network and graphlet adjacency, we apply for each pathway ten iterations of 5-fold cross-validation, where we predict which genes participate in it based on their

pathway-based graphlet eigencentrality (see Section: Annotation data, for details on the collected pathways). We evaluate prediction performance per pathway. That is, for each pathway and fold, we randomly hold out 20% of the genes known to participate in the pathway to form the positive examples in our test-set. The negative examples in the test-set are all genes in the full network that directly interact with one of the $m$ (i.e., 80%) of the remaining genes known to participate in the pathway.

**Prediction based on local pathway centrality.**    For each gene in the test set, we compute how central it would be in the pathway if it were to participate in it. That is, for each gene in the test set, we induce the nodes corresponding to the gene and the $m$ remaining genes known to participate in the pathway on the full network to define a local $(m + 1) \times (m + 1)$ dimensional adjacency matrix $P$, based on which we compute the local pathway centrality of the gene (see Section: Local pathway centrality). In this way, the centrality of each gene in the test set is based on local pathway topology, avoiding taking into account the 'noise' coming from interactions with nodes outside the pathway.

**Prediction based on global pathway centrality.**    For a given pathway, the underlying graphlet and a given fold, we compute the global pathway-based graphlet eigencentralities for all the genes in the test set (see Section Global pathway centrality). We consider genes with a higher global pathway-based graphlet eigencentrality to be more likely to be participating in the pathway.

**GeneMANIA.**    GeneMANIA is a supervised approach that uses a label propagation algorithm to predict gene annotations. We choose to compare against GeneMANIA as it: (1) is one of the few gene annotation predictors that, like our method, can be trained using only positive examples and (2) allows for sampling annotations from the pathway perspective rather than the gene perspective (i.e. for each pathway we hold out precisely 20% of the genes participating in it instead holding out the pathway annotations for 20% of all the genes, which would lead to pathways being unevenly sampled).

## Predicting cancer-related genes

We hypothesise that cancer-related genes play central roles in pathways and hence can be predicted based on their pathway based graphlet eigencentralities. For each pathway and graphlet adjacency, we directly use global or local graphlet eigencentrality to rank the genes participating in a given pathway, assuming that genes with a higher eigencentrality are more likely to be cancer-related (see Section Annotation data, for details on the collected pathways). For each pathway, we consider the set of known cancer driver genes participating in the pathway as the set of true positives. As here our approach is unsupervised (i.e. we do not use the information of which genes are known cancer drivers when computing pathway centralities), no cross-fold validation is needed.

## Evaluating prediction performance

We evaluate prediction performance on a per pathway and per graphlet adjacency basis using the area under the precision-recall curve (AUC-PR) and the area under the receiver operating characteristic curve (AUC-ROC), which are defined as follows.

For a given prediction, the *true positive rate* (TPR) is the number of correctly predicted true positives (i.e., the genes correctly predicted as part of the pathway or to be cancer driver genes) over all known true positives (i.e., all genes known to be part of the pathway or all cancer drivers in the pathway). The *false positive rate* (FPR) is defined as the number of genes falsely predicted as positive (i.e., the genes falsely predicted to be participating in a pathway or to be cancer driver genes). The ROC curve sets out the relationship between the TPR and FPR for

predicting pathway participation at various cut-offs. The AUC-ROC is used as a single number summary of the ROC curve, as a measure of prediction accuracy.

Similarly, for a given prediction, the *precision* is defined as the number of correctly predicted true positives (i.e., the number of genes correctly predicted to participate in the pathway, or the number of genes correctly predicted as cancer drivers) over the total number of genes in the prediction set (e.g., the known genes participating in the pathways and the genes they are directly connected with outside the pathway, or the all genes known to participate in the pathway). *Recall* is synonymous to the TPR, defined above. The precision-recall curve sets out the relationship between the precision and recall at various cut-offs. The area under the PR curve is then used as a single number summary of the precision-recall curve, as a measure of prediction accuracy.

To be able to identify the pathways or cancer mechanisms that are exceptionally well captured by a given graphlet adjacency, we define the normalized AUC-PR. For each graphlet adjacency and a given prediction task, we normalize the distribution of AUC-PR scores over all pathways by subtracting the median and dividing by the mean absolute deviation.

### Data

**Biological networks.** We create five unweighted and undirected networks based on three types of generic molecular interactions in human and baker's yeast (*S. cerevisiae*). We combine the experimentally validated protein-protein interactions (PPIs, validated using Two-hybrid or Affinity Capture based methods) from BioGRID version 3.5.178 [1] and PPIs from the Reactome Pathways [18] to form PPI networks, where nodes represent genes and edges represent physical interactions between their protein products. We collect gene co-expression (COEX) scores from COXPRESdb version 7.3 [2] to build COEX networks, where nodes represent genes and edges represent pairs of genes being co-expressed. We consider each gene to be co-expressed with its top 1% highest scoring co-expressed genes. For yeast, we collect experimentally validated genetic interactions (GIs) from BioGRID version 3.5.178 [1]. We exclude the human GI network from our analysis, as only a limited number of GIs is available. S1 Table in S1 File provides basic network statistics of these networks.

**Annotation data.** We collect pathway annotation data assigning genes to pathways, from the Reactome pathway ontology [18]. For each of our five molecular networks, we create a set of pathway networks by inducing the gene set of each pathway on the network. For each molecular network, we consider those pathways that, once induced on the full network, form a connected subgraph of a size of at least 10 and up to 100 nodes. We provide the distribution of pathway sizes for each of our molecular networks in S1 Fig in S1 File. The number of pathways considered per molecular network is summarized in S2 Table in S1 File.

We collect experimentally validated functional annotations from the Gene Ontology (i.e., evidence codes 'EXP', 'IDA', 'IPI', 'IMP', 'IGI', 'IEP'), that assign genes to biological process annotations (GO-BP), cellular component annotations (GO-CC) and molecular function annotations (GO-MF) [24].

We collect 586 cancer driver annotations from the intOGen database [25]. We consider a gene to be a cancer driver if it is a known cancer driver in at least one cancer type.

### Results and discussion

While this paper focuses on providing insight into the biology captured by the different graphlet adjacencies, in the appendix, we also investigate the agreement between our new graphlet eigencentrality measures and state-of-the-art centrality measures used in network biology (S1 File, Section: Comparing graphlet eigencentrality to other node centralities). There we show in

well-investigated model networks and our set of molecular networks that there are strong correlations between the different centralities, that depend on the network's topology. Despite this, we show in each of our model networks that there is some disagreement between the top 100 most central nodes based on the different graphlet eigencentralities, indicating their potential to capture complementary biological signal, which we investigate below.

## Graphlet adjacencies describe topologically and biologically distinct pathways

First, we validate that graphlet adjacencies can capture topological relationships between the nodes in a pathway by evaluating pathway participation prediction accuracy (Section: Graphlet adjacency captures pathway specific topology). Additionally, we validate that the pathways that are *described* by a given graphlet adjacency, i.e. the set of pathways for which we achieve the highest prediction accuracy considering a given graphlet adjacency, are topologically statistically significantly different from the rest of the pathways based on their degree distribution, average clustering coefficient and correlations between their graphlet counts (S1 File, Section: Pathways described by the same graphlet adjacency are topologically similar). Then, we find for each set of pathways, the set of model networks they are most similar to (e.g., Erdős-Rènyi random networks, scale-free networks, geometric networks, etc. See S1 File, Section: Linking pathways described by graphlet adjacencies to model networks). Then, we show that the pathways that are described by the same graphlet adjacency, share biological functional similarities that are different dependent on the graphlet adjacency considered (Section: Graphlet adjacencies describe complementary groups of functionally related pathways). We conclude this section with a case study, where we focus on the 'Receptor mediated mitophagy' pathway and explain why some graphlet adjacencies best capture the topological-functional relationships between nodes in the pathway (Section: Case study: Receptor mediated mitophagy).

Here we present results for the human PPI network. The results for our other molecular networks are presented in S1 File.

**Graphlet adjacency captures pathway specific topology.** We assess if graphlet adjacencies capture pathway topological signal by evaluating the performance of graphlet eigencentrality for the purpose of pathway participation prediction. In S17A Fig in S1 File. we observe that regardless of the underlying graphlet adjacency, our local approaches and GeneMANIA consistently perform better than random (AUC-ROC = 0.5), achieving median AUC-ROC scores higher than 0.7. Our global approach performs as by random when applied on graphlet adjacencies for $\widetilde{A_{G_1}}$, $\widetilde{A_{G_2}}$ and $\widetilde{A_{G_8}}$. Given that the ratio of positive examples in each test-set is only 0.15 on average, AUC-PR is a better measure for comparison. In S17B Fig in S1 File, we observe that our local approach outperforms our global approach, as well as GeneMANIA. To explain this result, we found that each pathway annotated gene participates in 6 pathways on average. Furthermore, on average, these 6 pathways are descendants of 2 (of the 23) different root nodes of the pathway ontology. This implies that from the perspective of the global network, pathways are intertwined, even functionally very distinct ones, making it harder to predict if a gene participates in a pathway or not. Our local approach, however, considers each pathway as an individual entity, disentangled from the rest of the network. This validates our intuition that, from the perspective of the pathway, all genes participating in it are important.

Next, to validate that different graphlet adjacencies best capture different sets of pathways, we compare the set of top-scoring pathways of each graphlet adjacency. We will be referring to the pathways for which we achieve the highest prediction accuracy considering a given graphlet adjacency as *described* by that graphlet adjacency. Formally, for each graphlet adjacency, we consider those pathways with a normalised AUC-PR score (see Section Evaluating prediction

performance) larger than 3 (in analogy to the 99.7% confidence interval for variables following a standard normal distribution) to be described by that graphlet adjacency. On average, 55 pathways are found to be described by a graphlet adjacency. By measuring the pairwise overlap between the set of pathways described by the different graphlet adjacencies, we find that the average of the Jaccard indices is 0.17. We conclude that graphlet adjacencies capture pathway topologies that are different and described by the underlying graphlet.

**Graphlet adjacencies describe complementary groups of functionally related pathways.** Having shown that graphlet adjacencies capture pathway topologies, we assess if any graphlet adjacency describes functionally similar pathways and compare the biological functions captured by different graphlet adjacencies.

To assess if a given graphlet adjacency captures similar pathways, we annotate each pathway with its second level *ancestors*, i.e. annotations in the second most general level of the pathway ontology, one step away from the root nodes and perform pathway set enrichment analysis (see S1 File, Section: Set enrichment analysis). The number of enriched ancestors in each set of pathways described by the different graphlet adjacencies are shown as bars in the bar chart in the top of Fig 2. We observe that each set of described pathways is enriched in at least three ancestor terms, meaning each set of described pathways has a comonality in terms of their pathway function. For instance, the set of pathways described by graphlet adjacency $\widetilde{A_{G_3}}$ is enriched in pathways related to 'Signaling by GPCR' (9 out of 59 pathways are descendants of this ancestor, adjusted p-value 2.23E−5), 'Transmission across Chemical Synapses' (9 out of 59 pathways are descendants of this ancestor, adjusted p-value 2.23E−5) and 'Platelet activation, signalling and aggregation' (6 out of 59 pathways are descendants of this ancestor, adjusted p-value 7.61E−23). Having established that the pathways described by the same graphlet adjacency are functionally related, we investigate if different graphlet adjacencies describe functionally different pathways. In the bottom of Fig 2, we present the Jaccard indices on the enriched ancestor terms, over all pairwise combinations of sets of pathways described by different graphlet adjacencies. The average of the Jaccard indices measuring the overlaps between the ancestors enriched in pathways described by two different graphlet adjacencies is 0.21, suggesting that different graphlet adjacencies capture functionally different pathways. For instance, while the ancestor annotation 'Signaling by GPCR' is enriched in the set of pathways described by graphlet adjacency $\widetilde{A_{G_3}}$, none of the 63 pathways described by graphlet adjacency $\widetilde{A_{G_6}}$ are annotated with it.

Analogously, we investigate if a given graphlet adjacency captures pathways that enriched in similar GO-terms (GO-BP, GO-CC and GO-MF, see Section Annotation data). We present the results in S1 File, Section: Graphlet adjacencies describe complementary groups of functionally related pathways. There, we observe that each set of pathways described by a given graphlet adjacency is highly enriched in pathways that are enriched in similar GO-BP, GO-CC and GO-MF terms, indicating that each graphlet adjacency describes biologically functionally similar pathways. We also observe that the average of the pairwise Jaccard indices, measuring the overlap in GO-terms enriched in pathways described by different graphlet adjacencies, is low: 0.51 for GO-BP, 0.47 for GO-CC and 0.49 for GO-MF. We present similar results for our other molecular networks. We conclude, with a few exceptions detailed in the appendix, that pathways described by a given graphlet adjacency are biologically functionally similar in terms of the ancestral, GO-BP, GO-CC and GO-MF terms in which they are enriched, and that these functional similarities depend on the graphlet adjacency. This suggests that each graphlet adjacency captures different pathway topology and function relationships.

**Case study: Receptor mediated mitophagy.** 'Receptor Mediated Mitophagy' (RMM) is a degradation process in the cell focused on the degradation of damaged mitochondria. We

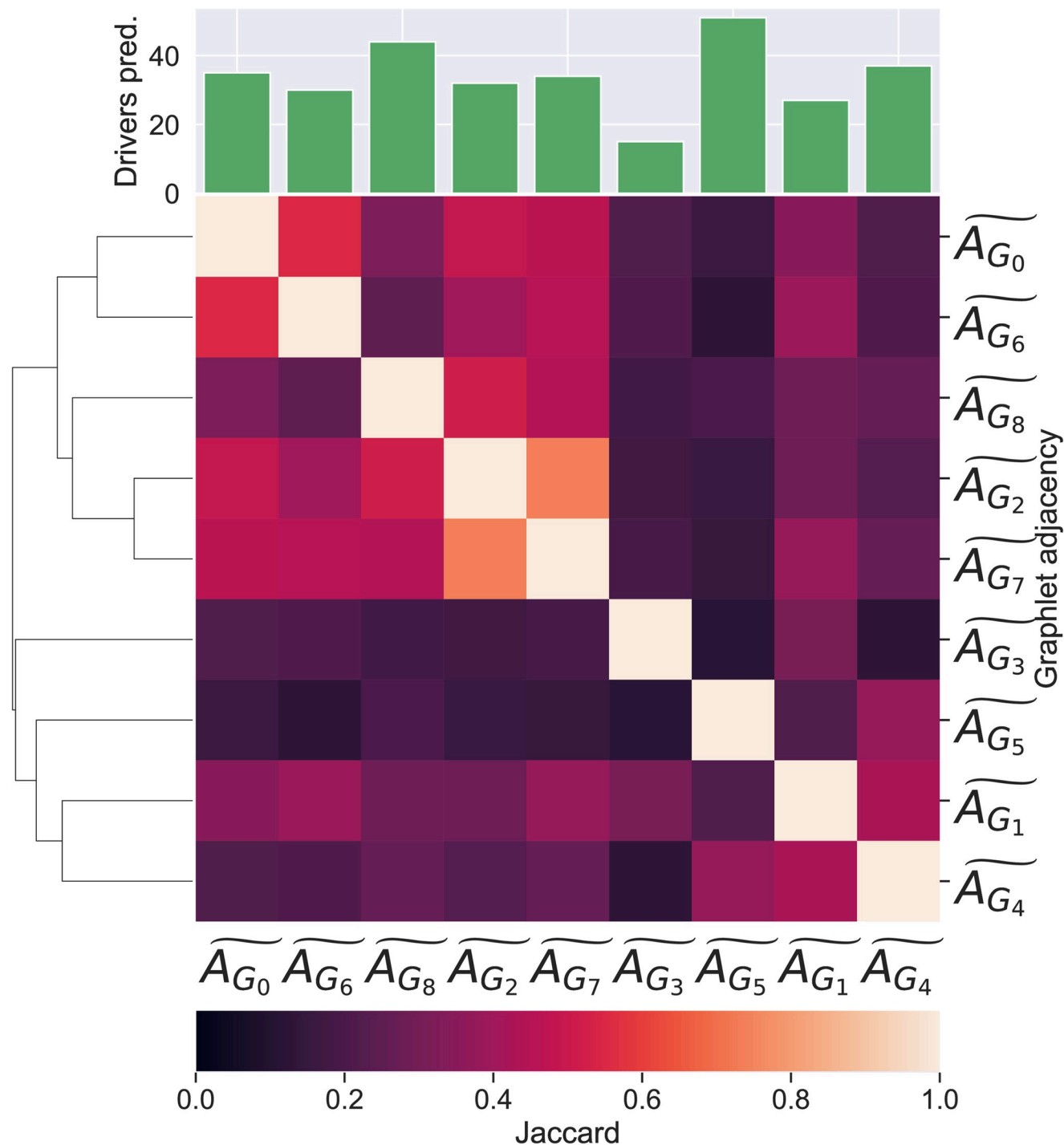

**Fig 2. Functional similarity between pathways described bydifferent graphlet adjacencies.** (Bottom) A clustered heat map of the Jaccard similarity indices between the sets of ancestor annotations enriched in the sets of pathways described by different types of graphlet adjacencies. (Top) A bar-chart indicating the number of ancestor annotations enriched in the pathways described by each corresponding graphlet adjacency.

found the pathway to be highly described by graphlet adjacency $\widetilde{A_{G_6}}$ (normalised AUC-PR 5.98) and not described by $\widetilde{A_{G_0}}$ (normalised AUC-PR 0.04), and will be focusing on this pathway to explain why some graphlet adjacencies better capture some pathways than others.

In Fig 3 we show the spring embedding of RMM based on normalised graphlet adjacencies $\widetilde{A_{G_0}}$ and $\widetilde{A_{G_6}}$. For graphlet adjacency $\widetilde{A_{G_0}}$, the RMM pathway is composed of two densely connected modules, the control mechanism (genes CSNK2A, CSNK21, CSNK2B, SRC) and the phagophore formation process (genes ATG5, ATG12, MAP1LCA, MAP1LCB, ULK1), which interact through a single hub gene, FUNDC1. This is unfavourable for prediction, as a gene would be predicted to be part of the pathway if it is densely connected with just one of the two clusters. Graphlet adjacency $\widetilde{A_{G_6}}$, however, does capture the fact that, through hub node FUNDC1, all the genes in the control mechanism and the phagophore formation process are functionally related (i.e. executing the RMM process), as both groups of genes are now highly

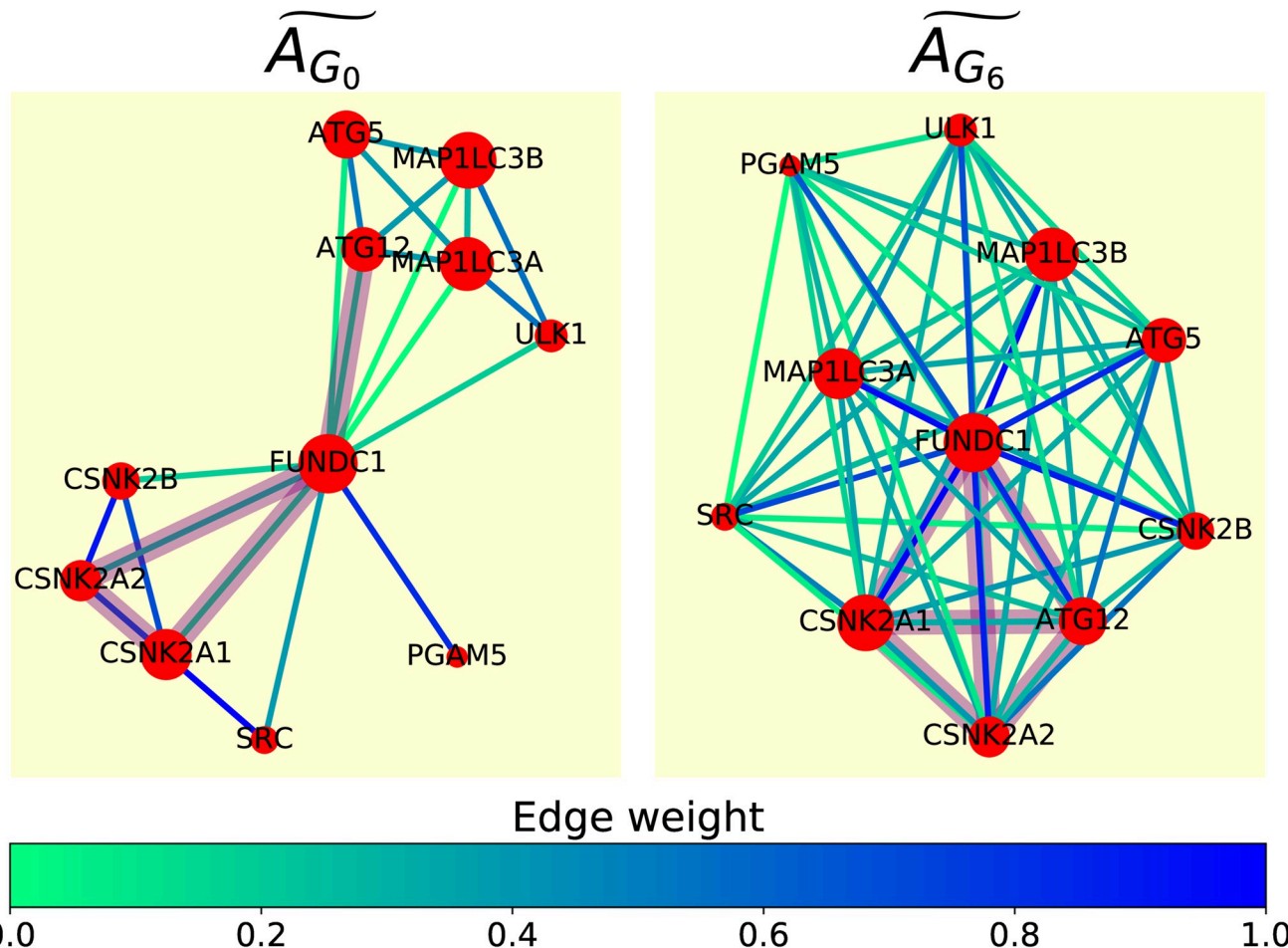

**Fig 3. Graphlet adjacency $\widetilde{A_{G_6}}$ captures RMM functional organisation.** Spring embedding of RMM based on normalised graphlet adjacency $\widetilde{A_{G_0}}$ (left) and $\widetilde{A_{G_6}}$ (right), where nodes represent genes (red) and edges represent weighted normalised graphlet adjacency (see legend). Graphlet $G_6$ is indicated in purple in the spring embedding based on $G_0$, connecting genes CSNK2A1, CSNK2A2, FUNDC1 and ATG12. The subnetwork obtained by inducing these same nodes is also indicated in purple in the spring embedding based on graphlet adjacency $\widetilde{A_{G_6}}$. Although only connected via FUNDC1 when considering regular adjacency, ATG12 is directly connected to CSNK2A1 and CSNK2A2 in the spring embedding based on graphlet adjacency $\widetilde{A_{G_6}}$, illustrating how graphlet adjacencies capture functionally relevant indirect relationships between nodes.

connected. This also better captures the pathway from a topological perspective, as genes predicted to be part of the pathway would have to interact (in the form of graphlet $G_6$) with all pathway members. We conclude that graphlet adjacency allows us to describe the functional organization of the pathway.

## Graphlet adjacencies capture complementary cancer mechanisms

Here, we illustrate how graphlet eigencentralities enable us to relate specific local wiring patterns of genes in a pathway with their individual biological function. We focus on predicting cancer driver genes. We first verify that cancer-related genes play central roles in pathways (Section: Graphlet eigencentrality captures the central roles of cancer related genes). Then, we show that the set of cancer driver genes recognised for playing central roles in pathways are different based on the graphlet adjacency considered (Section Different graphlet adjacencies uncover different cancer-related genes). To explain this, we illustrate it with a case study, where we show why graphlet adjacency $\widetilde{A_{G_6}}$ best captures the central roles of cancer driver genes, TP53 and RB1, in the 'Formation of Senescence-Associated Heterochromatin Foci' pathway (Section: Case study: Formation of Senescence-Associated Heterochromatin Foci (FSAHF)).

In this part of the study, we consider all non-disease specific pathways in Reactome (see S1 File, Section: Basic network statistics). We present results for the human PPI network, with the corresponding results for the human COEX network presented in the appendix.

**Graphlet eigencentrality captures the central roles of cancer related genes.** We assess if cancer driver genes tend to have central positions in pathways by performing the following analysis. For each pathway, we predict its genes to be cancer-related according to their pathway centrality. We consider the set of cancer driver genes provided by intOGen as our set of true positives (Section: Annotation data). The results are presented in S38 Fig in S1 File. We observe that both our local and our global approach perform better than random (AUC-ROC = 0.5), as median AUC-ROC scores over all pathways are typically higher than 0.60, regardless of the underlying graphlet adjacency. We observe that global graphlet eigencentrality consistently outperforms local graphlet eigencentrality. To explain this, we perform a Mann-Whitney U (MWU) test comparing the distribution of the number of pathways that each cancer driver genes occurs in, with the distribution of the number of pathways that each non-cancer driver gene occurs in. We find that, in the human PPI network, cancer driver genes occur on average in almost twice as many pathways as non-cancer driver genes (10.56 compared to 6.07), which is statistically significantly different with a p-value of 5.19E–20. Therefore, while cancer genes tend to have a central role in pathways (as indicated by our local graphlet eigencentralities), our results also suggest that they play a more critical role in the crosstalk between the pathways regulating the cell (as indicated by global graphlet eigencentralities). These results are in line with the existing literature, as cancer driver genes have been shown to have a statistically significantly higher betweenness centrality than other genes in the PPI network [26]. Looking for specific examples of cancer driver genes playing a role in cancer through crosstalk, we find, for instance, that the crosstalk between cancer driver STAT3 and the p53/RAS signaling pathway controls cancer cell metastasis [27]. Similarly, crosstalk between p53 and the IGF-1R/AKT/mTORC1 pathway can lead to chemo resistance [28].

We consider the study of how cancer related genes interact between pathways as future work and will be focussing on illustrating how graphlet eigencentralities capture pathway mechanisms within pathways.

**Different graphlet adjacencies uncover different cancer-related genes.** First, we focus on those pathways *described* by central cancer driver genes, i.e. those pathways for which we

achieve a normalized AUC-PR score larger than 3 applying local graphlet eigencentralities. Additionally, we determine for each pathway a set of *correctly predicted cancer-related genes*. For each pathway, we determine the threshold such that the F1 score for predicting cancer drivers in that pathway is maximal and consider all the known cancer driver genes with a centrality score higher than this threshold as correctly predicted cancer-related genes. In Fig 4, we show the pairwise Jaccard indices between the sets of correctly predicted genes uncovered based on different graphlet adjacencies. With an average Jaccard index of 0.30, we conclude that different graphlet adjacencies describe the role in cancer of different sets of cancer related genes.

**Case study: Formation of Senescence-Associated Heterochromatin Foci (FSAHF).** The formation of senescence-associated heterochromatin foci (FSAHF), contributes to senescence (permanent interruption of cell division) by repressing the expression of proliferation-promoting genes through reorganisation of chromatin [29]. Cellular senescence plays a vital role in permanently restricting the propagation of damaged and defective cells and forms a natural tumour-suppressor mechanism. We found the cancer mechanism in the FSAHF pathway to be described by graphlet adjacency $\widetilde{A_{G_6}}$ (normalised AUC-PR 3.2) and poorly described by $\widetilde{A_{G_0}}$ (normalised AUC-PR −0.56). We will be focusing on this pathway to explain how graphlet adjacencies can capture different cancer mechanisms in pathways.

In Fig 5, we show the spring embedding of the SAHF formation pathway-based on normalised graphlet adjacency $\widetilde{A_{G_0}}$ and $\widetilde{A_{G_6}}$. From the perspective of graphlet adjacency $\widetilde{A_{G_0}}$, cancer drivers RB1 and TP53 do not play a central role in this pathway, as they appear peripheral to the other nodes in the pathway. The mediating role of TP53 and RB1 trough hub node HMGA2 is well captured by graphlet adjacency $\widetilde{A_{G_6}}$, connecting them with all nodes in the pathway. Additionally, through literature curation, we find that HMGA2, the most central node in the pathway according to graphlet adjacency $\widetilde{A_{G_6}}$ and predicted as cancer-related in Section: Different graphlet adjacencies uncover different cancer-related genes, is also a driver of tumour metastasis [30]. Lastly, it should be noted that within this pathway, nodes UBN1, ASF1A, TP53 touch graphlet $G_0$ the most (i.e. have the highest degree) and nodes EP400, RB1 and H1–0 touch graphlet $G_6$ the most (i.e. have the highest graphlet degree for graphlet $G_6$). This means that the central roles of TP53 and RB1 trough hub node HMGA2 could not have been captured neither by using the simple degree centrality, nor by using their graphlet degree centrality for graphlet $G_6$. We conclude that graphlet eigencentrality enables considering different notions of the centrality of genes in pathways, allowing the capturing of different functional roles of genes in pathways.

## Conclusion

In this paper, we introduce graphlet eigencentrality, allowing us to capture different notions of the centrality of nodes in a network. We apply it on measuring the centrality of genes in pathways, enabling a detailed investigation of how different graphlet adjacencies capture different biological functions. We apply our method at two levels: from the local pathway perspective or the global network perspective.

We apply our graphlet eigencentralities to identify pathways described by different graphlet adjacencies, i.e. all genes participating in a pathway are also be important from the topological perspective. To do so, we use our graphlet eigencentralities to predict which genes belong to a given pathway, considering the pathways for which we achieve the highest prediction accuracies as being described by that graphlet adjacency. We find that local pathway-based graphlet eigencentralities well predict which genes participate in a given pathway, outperforming state

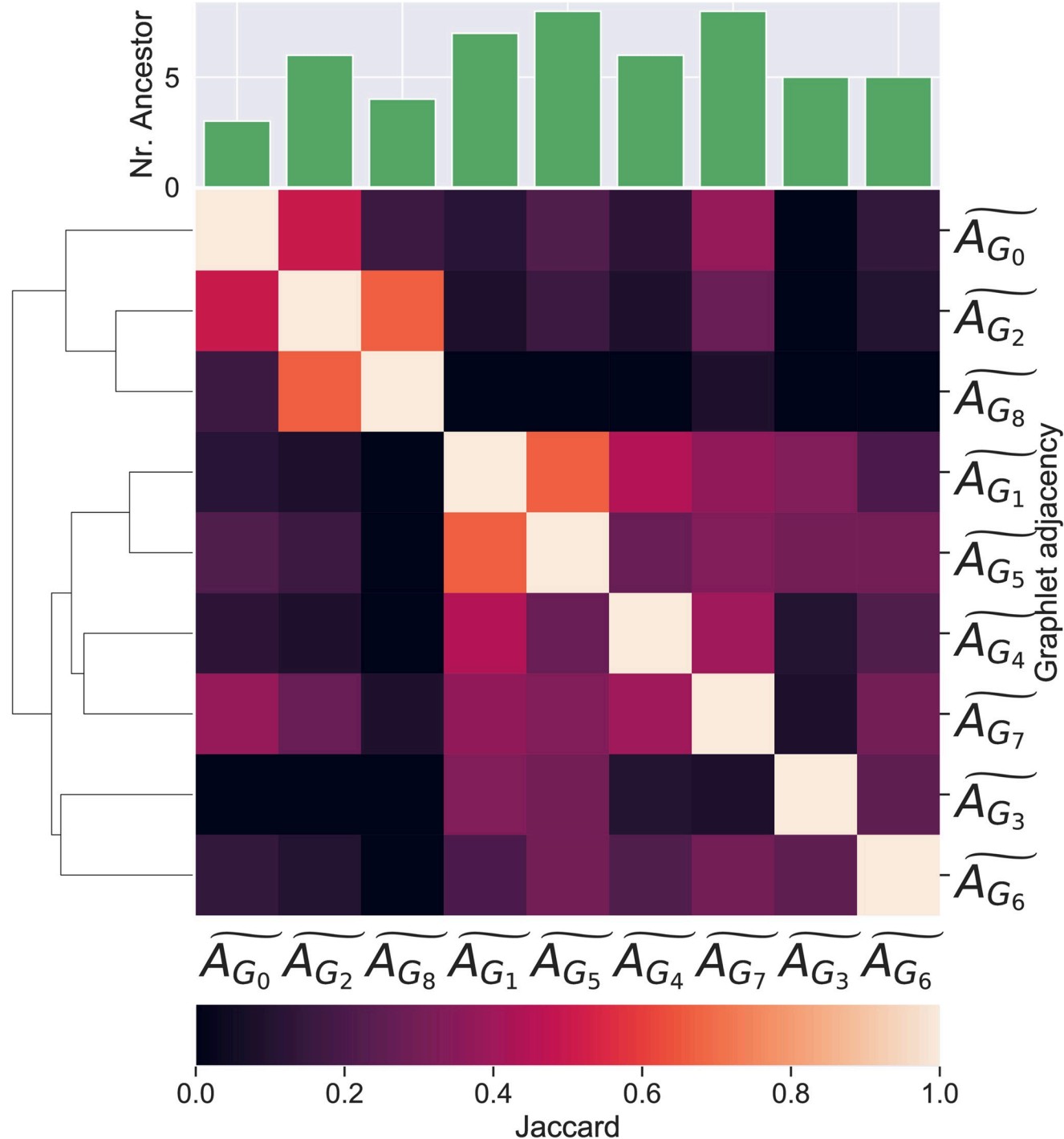

**Fig 4. The overlap between correctly predicted cancer genes in pathways described by central cancer genes based on different graphlet adjacencies.** A clustered heat map of the Jaccard similarity indices between the sets of correctly predicted cancer genes found in pathways described by central driver genes based on different types of graphlet adjacencies. At the top, the bar-chart indicates the number of correctly predicted genes corresponding to each graphlet adjacency.

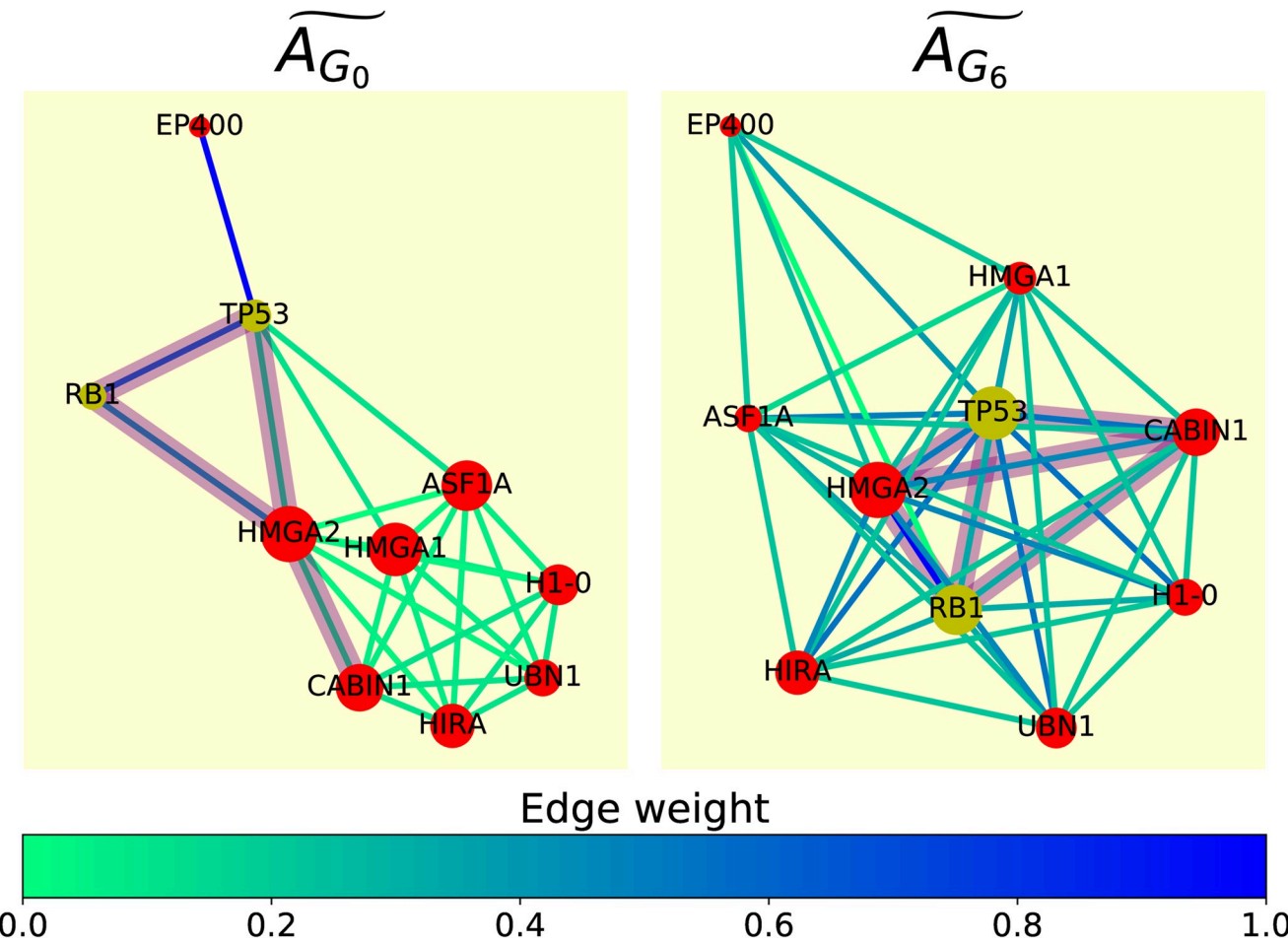

**Fig 5. Graphlet adjacency $\widetilde{A_{G_6}}$ captures centrality of cancer driver genes in the FSAHF pathway.** Spring embedding of FSAHF based on normalised graphlet adjacency $\widetilde{A_{G_0}}$ (left) and $\widetilde{A_{G_6}}$ (right), where nodes represent non-cancer-related genes (red) and known cancer driver genes RB1 and TP53 (yellow), and edges represent weighted normalised graphlet adjacency (see legend). Graphlet $G_6$ is indicated in translucent purple in the spring embedding based on $G_0$, connecting genes RB1, TP53, HMGA2 and CABIN1. The subnetwork obtained by inducing these same nodes is also indicated in translucent purple in the spring embedding based on graphlet adjacency $\widetilde{A_{G_6}}$. Although only connected via HMGA2 when considering regular adjacency, TP53 and RB1 are directly connected to CABIN1 in the spring embedding based on graphlet adjacency $\widetilde{A_{G_6}}$, illustrating how graphlet adjacencies can capture functionally relevant indirect relationships between nodes.

of the art predictor GeneMANIA (validating our approach) and our global approach. To explain this result, we show that pathways, even when functionally unrelated, show large amounts of overlap. As our local approach considers each pathway as an individual entity disentangled from the full network, it is able to best capture the topological essence of a pathway. We go on to show that pathways that are described by a given graphlet adjacency are biologically functionally similar in terms of the ancestral, GO-BP, GO-CC and GO-MF terms in which they are enriched, and that these functional similarities depend on the graphlet adjacency. We illustrate these results by a case study of the 'Receptor mediated mitophagy' pathway, where we show how graphlet adjacency $\widetilde{A_{G_6}}$ captures the hub-role of FUNDC1, allowing us to capture the functional organisation of the pathway.

Secondly, we apply our graphlet eigencentrality at predicting cancer-related genes in pathways. We observe that global graphlet eigencentrality consistently outperforms local graphlet

eigencentrality. To explain this result, we show that cancer driver genes participate in statistically significantly more pathways than non-cancer-related genes. Therefore, while cancer genes tend to have central roles in pathways (as indicated by our local graphlet eigencentralities), our results also suggest that they play a more essential role in the crosstalk that occurs between pathways to regulate the cell (as indicated by our global graphlet eigencentralities). This is a key insight, as it indicates that pathway-focused approaches for studying cancer should focus on the interactions between pathways, although most current state-of-the-art approaches focus on their individual differential expression or rewiring. Additionally, we show that by considering pathway centrality based on different graphlets, we can uncover complementary sets of cancer genes. We illustrate these results by a case study of the FSAHF pathway, where we show how graphlet adjacency, unlike regular adjacency, captures the central roles of cancer driver genes, RB1 and TP53. We conclude that graphlet eigencentralities allow us to capture different functional roles of genes in and between pathways.

Finally, our graphlet eigencentralities can be applied to study diseases outside cancer. For instance, it has been shown that rare-disease genes are characterised by a high degree and a high betweenness centrality in the PPI network [31]. Further, the study of the centrality of nodes is not limited to biology, making our graphlet eigencentralities applicable in many disciplines that use networks as models, including physics, social sciences and economics.

## Supporting information

**S1 File. Supplementary methods, figures and tables.**
(PDF)

## Author Contributions

**Conceptualization:** Sam F. L. Windels.

**Data curation:** Sam F. L. Windels.

**Formal analysis:** Sam F. L. Windels.

**Funding acquisition:** Nataša Pržulj.

**Investigation:** Sam F. L. Windels, Noël Malod-Dognin.

**Methodology:** Sam F. L. Windels.

**Supervision:** Noël Malod-Dognin, Nataša Pržulj.

**Writing – original draft:** Sam F. L. Windels.

**Writing – review & editing:** Noël Malod-Dognin, Nataša Pržulj.

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
