## [Decision Letter · Decision Letter 0]

20 Jul 2021

PONE-D-21-17376

Graphlet eigencentralities capture novel central roles of genes in pathways

PLOS ONE

Dear Dr. windels,

Thank you for submitting your manuscript to PLOS ONE. After careful consideration, we feel that it has merit but does not fully meet PLOS ONE’s publication criteria as it currently stands. Therefore, we invite you to submit a revised version of the manuscript that addresses the points raised during the review process.

You will see that while the reviewers are persuaded of the importance and quality of your study, they have raised several points that require consideration and minor revision.  In particular, Reviewer 2 suggests comparison of your method with simpler topological metrics and additional discussion of applicability of this method to more diverse disease**.**

We look forward to receiving your revised manuscript.

Kind regards,

Katherine James, Ph.D.

Academic Editor

PLOS ONE

Journal Requirements:

Reviewers' comments:

Reviewer's Responses to Questions

**Comments to the Author**

1. Is the manuscript technically sound, and do the data support the conclusions?

Reviewer #1: Yes

Reviewer #2: Yes

2. Has the statistical analysis been performed appropriately and rigorously? 

Reviewer #1: Yes

Reviewer #2: Yes

3. Have the authors made all data underlying the findings in their manuscript fully available?

Reviewer #1: Yes

Reviewer #2: Yes

4. Is the manuscript presented in an intelligible fashion and written in standard English?

Reviewer #1: Yes

Reviewer #2: Yes

5. Review Comments to the Author

Reviewer #1: The paper "Graphlet eigencentralities capture novel central roles of genes in pathways." shows an interesting methodology for analyzing the relevance of nodes in complex biochemical networks. It shows how to take advantage in a deeper way of the information in the graphlet adjacency matrix to reveal the most important nodes in a network in terms of its biological functions. In particular, the paper focuses its attention on analyzing networked data coming from biochemistry experiments, e.g., protein-protein interaction networks, metabolic networks, and genomic networks.

The main idea is to use all topological information in the graphlets, infer relevance and biological function based on how nodes participate in some graphlets. In addition, the authors adopt the heuristics of eigencentrality. In few words, it establishes that a node is important if the neighbors of such node are also important.

The information that graphlet adjacency matrix offers can enrich the 'omics' networked data and give insights into the function of the nodes in a biomolecular network.

This approach is related to the one developed in [1], where the networks' information is enriched using similarities between neighborhoods through various metrics. In this sense, I recommend mentioning the analogies of both approaches when exploiting topological information more deeply. Moreover, both use the heuristic of self-centrality as a methodology for calculating the relevance of a node. In addition, I recommend to add some reviews about centrality in complex networks such as [2] or [3] to complete the literature on the topic of centrality measures. In addition, eigencentrality is due to [4] but instead is cited a general book that loses the author's original inspiration. I recommend citing the original author of the method.

Although the concept of centrality itself has not yielded universal results for all types of networks, it is because centralities are typically inspired in some definition of importance that can not be applied to all kinds of networks. In this sense, eigencentrality based on graphlets seems to be a good way to capture different types or notions of centrality behind nodes.

In general, the paper is quite clear and interesting, the methodology is mathematically correct, and its applications are highly relevant for biochemists and sociologists, physicists, computer scientists, among many other disciplines, since the methodology is universal.

Before adding the complementary references, I think that the work could be published.

[1] Alvarez-Socorro, A. J., Herrera-Almarza, G. C., & González-Díaz, L. A. (2015). Eigencentrality based on dissimilarity measures reveals central nodes in complex networks. Scientific reports, 5(1), 1-10.

[2] Xiaolong, R., & Linyuan, L. (2014). Review of ranking nodes in complex networks. Chinese Science Bulletin, 59(13), 1175-1197.

[3] Landherr, A., Friedl, B., & Heidemann, J. (2010). A critical review of centrality measures in social networks. Business & Information Systems Engineering, 2(6), 371-385.

[4] Bonacich, P. (1972). Factoring and weighting approaches to status scores and clique identification. Journal of mathematical sociology, 2(1), 113-120.

Reviewer #2: Overview:

This manuscript builds up on previous work published by the authors regarding graphlets and node adjacency in biological networks. Here, they introduce the concept of graphlet-based eigencentralities to identify pathways and cancer mechanisms described by different graphlet adjacencies. Interestingly, through graphlet eigencentralities the authors show that specific graphlet adjacencies describe functionally similar biological processes and also predict central cancer driver genes. There is a load of both sophisticated and complicated work behind the manuscript and the provided Supplement is crucial to allow the reader to better understand and follow the concepts. It would be very interesting if graphlet theory as described in this manuscript could predict implicated genes in other diseases as well, especially rare ones. Also, since the described theory is complicated, a future direction for the authors would be to create a software suite that automates the described pipeline to extract knowledge.

Major Comments:

1. Supplement (page 11): “different graphlet eigencentralities are positively correlated with each other and most existing centrality measures”. I was wondering what results the simple degree metric (instead of the sophisticated graphlet eigencentralities) would yield, regarding the cancer central gene case studies. Have the authors tried to compare these? For example, what is the degree of HMGA2 in the underlying network regarding the FSAHF case study? In a nutshell: how do graphlet eigencentralities outperform simple topological metrics in the topic of uncovering key implicated players in perturbed pathways?

2. Could the detection of key genes through graphlet eigencentralities work on other diseases? Or is there something special regarding cancer pathways (such as the cross-talk among them) that allows the global pathway centralities to predict these key genes only in this scenario? The authors could include a third case study with a non-cancer disease pathway to figure this out.

Minor Comments:

1. The introduction has been structured sufficiently, explaining all concepts relative to the manuscript and leading the reader to the problem/objective which is presented. Regarding the three approaches for inferring gene functions in networks; I am not sure Centrality (and other topological metrics such as clustering coefficient and degree) is distinguishable from Topology. Maybe rename the third category specifically into “graphlet-based”, since this has already been established by the authors previous publications.

2. Figure 1: Need to sharpen resolution.

3. Figure 1-C: Refer to it in text after (3) is described.

4. “(Section )” is written a number of times across the manuscript. Is this supposed to be a link? Else, write down the respective sections.

5. Explain λ (line 131) a bit more; λ are eigenvalues for which a non-zero eigenvector solution exists.

6. Maybe use another letter to describe the extended definition of adjacency instead of AGi, which is used for the standard graphlet adjacency matrix to avoid confusion. As I understand (line 148), cGiuv is the value of the standard graphlet adjacency AGi(u,v), correct?

7. Is there a brief explanation on why the degree is raised to the power of -1/2 in (5)? Could 1/degree also work here?

8. (line 212) comma typo: “that ,like” -> “that, like”

9. (lines 214-215) What is the difference between “20% of the genes of each pathway” and “20% of the pathway annotated genes”? Maybe rephrase to better clarify.

10. (line 219) “For each pathway”: Write down which pathways were used here.

11. Supplement (Page 44): “Here validate”: typo: “Here we validate”

12. Number the sections in the main manuscript as referred to in the Supplement (e.g., “see Section 2.7.1 in the main paper”), or write the section names 9instead of numbers) in the Supplement.

13. (line 316) Typo: “for the purpose pathway participation” -> “for the purpose of pathway participation”

14. (line 317-319): “In Supplementary Figure 17-A we observe that regardless of the underlying graphlet adjacency, our local approaches and GeneMANIA consistently perform better than random”. Add in parentheses “red line” or “AUC-ROC = 0.5”, to better clarify.

15. (line 339) Typo: fonund -> found

16. (line 409) Typo: enables -> enable

17. (line 431) explain what “MWU” stands for (Mann-Whitney U test). This is written in the Supplement but the main manuscript should be complete by itself.

18. Figure 5: color coding stated on the figure legend is not correct. All nodes appear red instead of red/yellow “nodes represent non-cancer-related genes (red) and known cancer driver genes RB1 and TP53 (yellow)”. Also, node ASF1A on the left figure should be moved, to not obscure the TP53-RB1 connection.

19. Conclusion (lines 517-519): “Additionally, we show that cancer genes that can be uncovered by their pathway centrality are different depending on the graphlet eigencentrality”. Different as in what? Described mechanisms? Please, rephrase to clarify.

20. Conclusion (line 524): Delete “using networkss”.

6. PLOS authors have the option to publish the peer review history of their article (what does this mean?). If published, this will include your full peer review and any attached files.

Reviewer #1: No

Reviewer #2: No

---

## [Author Response · Author response to Decision Letter 0]

19 Aug 2021

Response to the reviewers

Reviewer #1: 

Comment 1: … This approach is related to the one developed in [1], where the networks' information is enriched using similarities between neighborhoods through various metrics. In this sense, I recommend mentioning the analogies of both approaches when exploiting topological information more deeply. Moreover, both use the heuristic of self-centrality as a methodology for calculating the relevance of a node. In addition, I recommend to add some reviews about centrality in complex networks such as [2] or [3] to complete the literature on the topic of centrality measures. In addition, eigencentrality is due to [4] but instead is cited a general book that loses the author's original inspiration. I recommend citing the original author of the method. …

Before adding the complementary references, I think that the work could be published.

[1] Alvarez-Socorro, A. J., Herrera-Almarza, G. C., & González-Díaz, L. A. (2015). Eigencentrality based on dissimilarity measures reveals central nodes in complex networks. Scientific reports, 5(1), 1-10.

[2] Xiaolong, R., & Linyuan, L. (2014). Review of ranking nodes in complex networks. Chinese Science Bulletin, 59(13), 1175-1197.

[3] Landherr, A., Friedl, B., & Heidemann, J. (2010). A critical review of centrality measures in social networks. Business & Information Systems Engineering, 2(6), 371-385.

[4] Bonacich, P. (1972). Factoring and weighting approaches to status scores and clique identification. Journal of mathematical sociology, 2(1), 113-120.

Response: 

As suggested by the reviewer, we add [1] as a related eigencentrality based measure. In particular, we add the following paragraph (lines 140-143 of page 4 in the revised manuscript): “Many variations of eigencentrality exist. For instance, the Katz centrality generalises the eigencentrality to directed networks [5]. The contribution centrality extends the eigencentrality by amplifying a node’s centrality if it serves as a hub node connecting densely connected parts of the network [1].”.

As suggested by the reviewer, we add [3] as a recommended review paper on centrality measures and [4] as a reference for eigencentrality. We did not add [2] as it is written in Chinese. 

Additional reference:

[5] Katz L. A new status index derived from sociometric analysis. Psychometrika. 1953;18(1):39-43.

Reviewer #2: 

Major Comments:

Comment 1: Supplement (page 11): “different graphlet eigencentralities are positively correlated with each other and most existing centrality measures”. I was wondering what results the simple degree metric (instead of the sophisticated graphlet eigencentralities) would yield, regarding the cancer central gene case studies. Have the authors tried to compare these? For example, what is the degree of HMGA2 in the underlying network regarding the FSAHF case study? In a nutshell: how do graphlet eigencentralities outperform simple topological metrics in the topic of uncovering key implicated players in perturbed pathways?

Response: In our FSAHF case study, we show how graphlet adjacency for graphlet G6 captures the central roles of cancer drivers TP53 and RB1. These would not have been uncovered neither based on their simple degree centrality, nor based on their graphlet degree centrality for graphlet G6 (i.e. the number of times a node touches graphlet G6). We updated the text to highlight this (lines 497-502 of page 13 in the revised manuscript): “Lastly, it should be noted that within this pathway, nodes UBN1, ASF1A, TP53 touch graphlet G0 the most (i.e. have the highest degree) and nodes EP400, RB1 and H1-0 touch graphlet G6 the most (i.e. have the highest graphlet degree for graphlet G6). This means that the central roles of TP53 and RB1 trough hub node HMGA2 could not have been captured neither by using the simple degree centrality, nor by using their graphlet degree centrality for graphlet G6.”.

Comment 2: Could the detection of key genes through graphlet eigencentralities work on other diseases? Or is there something special regarding cancer pathways (such as the cross-talk among them) that allows the global pathway centralities to predict these key genes only in this scenario? The authors could include a third case study with a non-cancer disease pathway to figure this out.

Response: We agree with the reviewer that our work opens up questions with respect to diseases outside cancer. However, given the length of the current manuscript and the supplement, we add the reviewer’s suggestion as proposed future work (lines 544-546 of page 14 in the revised manuscript): “Finally, our graphlet eigencentralities can be applied to study diseases outside cancer. For instance, it has been shown that rare-disease genes are characterised by a high degree and a high betweenness centrality in the PPI network [30].”

Minor Comments:

Comment 1: The introduction has been structured sufficiently, explaining all concepts relative to the manuscript and leading the reader to the problem/objective which is presented. Regarding the three approaches for inferring gene functions in networks; I am not sure Centrality (and other topological metrics such as clustering coefficient and degree) is distinguishable from Topology. Maybe rename the third category specifically into “graphlet-based”, since this has already been established by the authors previous publications.

Response: We followed the suggestion of the reviewer and renamed this section to “graphlet-based” (lines 34-36 of page 2 in the revised manuscript).

Comment 2: Figure 1: Need to sharpen resolution.

Response: The figure is now rendered at 300X300 DPI, as per the PLOS ONE style guide.

Comment 3: Figure 1-C: Refer to it in text after (3) is described.

Response: We now refer to Figure 1-C as an illustration for graphlet adjacency defined in formula (3) in the following sentence (line 155 of page 5 in the revised manuscript): “We illustrate A_G0 and A_G1 in Figure 1-C.”

Comment 4: “(Section )” is written a number of times across the manuscript. Is this supposed to be a link? Else, write down the respective sections.

Response: We thank the reviewer for reporting this compilation issue. We now consistently refer to sections using the name of the section, as per the PLOS ONE style guide. 

Comment 5: Explain λ (line 131) a bit more; λ are eigenvalues for which a non-zero eigenvector solution exists.

Response: We agree with the reviewer and we added the suggestion to the text (lines 134-136 of page 4 in the revised manuscript): “From this, it is clear that c is an eigenvector of A and λ is an eigenvalue for which a non-zero eigenvector solution exists; hence the name `eigencentrality'.”

Comment 6: Maybe use another letter to describe the extended definition of adjacency instead of AGi, which is used for the standard graphlet adjacency matrix to avoid confusion. As I understand (line 148), cGiuv is the value of the standard graphlet adjacency AGi(u,v), correct?

Response: In our naming scheme, A_(G_i ) stands for graphlet adjacency for graphlet G_i. As graphlet G_0is an edge, A_(G_0 )is equivalent to the standard adjacency matrix. We modified the text to make this more clear. 

In particular, we modified the following paragraph which in the old version of the manuscript said: 

“… where C_uv^(G_i )is equal to the number of times the nodes u and v are graphlet adjacent w.r.t. graphlet G_i and θ_(G_i ) is equal to the number of nodes in graphlet G_i minus 1. Note that graphlet adjacency matrix A_(G_0 ), is equivalent to the standard graph adjacency matrix.” ,

and which now reads (lines 152-154 of page 5 in the revised manuscript): 

“… where C_uv^(G_i )is equal to the number of times the nodes u and v simultaneously touch graphlet G_i and θ_(G_i ) is a scaling constant equal to the number of nodes in graphlet G_i minus 1. Note that graphlet adjacency matrix A_(G_0 ), is equivalent to the standard graph adjacency matrix.”

Comment 7: Is there a brief explanation on why the degree is raised to the power of -1/2 in (5)? Could 1/degree also work here?

Response: We added the following sentence to add the intuition behind symmetric normalisation (lines 161-162 of page 5 in the revised manuscript): “Intuitively, the symmetric normalisation rescales the weight of a given edge relative to its importance to both nodes involved.”

Comment 8: (line 212) comma typo: “that ,like” -> “that, like”

Response: We amended the typo. The sentence now reads (line 219 of page 6 in the revised manuscript): “We choose to compare against GeneMANIA as it: (1) is one of the few gene annotation predictors that, like our method, can be trained using only positive examples and (2) …”. 

Comment 9: (lines 214-215) What is the difference between “20% of the genes of each pathway” and “20% of the pathway annotated genes”? Maybe rephrase to better clarify.

Response: We agree with the reviewer. The sentence now reads as follows (lines 218-223 of pages 6-7 in the revised manuscript): “We choose to compare against GeneMANIA as it: (1) … and (2) allows for sampling annotations from the pathway perspective rather than the gene perspective (i.e. for each pathway we hold out precisely 20% of the genes participating in it instead holding out the pathway annotations for 20% of all the genes, which would lead to pathways being unevenly sampled).”

Comment 10: (line 219) “For each pathway”: Write down which pathways were used here.

Response: We agree with the reviewer and now refer to “Section: Annotation data,” where more details about the pathways collected are described (lines 229-230 of page 7 in the revised manuscript).

Comment 11: Supplement (Page 44): “Here validate”: typo: “Here we validate”

Response: We amended this typo (page 45 in the revised supplement). The sentence now reads as follows: “Here we validate that the pathways that are described by the same graphlet adjacency are statistically significantly topologically similar”.

Comment 12: Number the sections in the main manuscript as referred to in the Supplement (e.g., “see Section 2.7.1 in the main paper”), or write the section names 9instead of numbers) in the Supplement.

Response: We now consistently refer to sections using the name of the section, as per the PLOS ONE style guide.

Comment 13: (line 316) Typo: “for the purpose pathway participation” -> “for the purpose of pathway participation”

Response: This typo was amended. The sentence now reads as follows (lines 327-329 of page 9 in the revised manuscript ).“We assess if graphlet adjacencies capture pathway topological signal by evaluating the performance of graphlet eigencentrality for the purpose of pathway participation prediction.”

Comment 14: (line 317-319): “In Supplementary Figure 17-A we observe that regardless of the underlying graphlet adjacency, our local approaches and GeneMANIA consistently perform better than random”. Add in parentheses “red line” or “AUC-ROC = 0.5”, to better clarify.

Response: We agree with the reviewer and amend the sentence as suggested. The sentence now reads as follows (lines 329-331 of page 9 in the revised manuscript): “In Supplementary Figure 17-A we observe that regardless of the underlying graphlet adjacency, our local approaches and GeneMANIA consistently perform better than random (AUC-ROC=0.5), achieving median AUC-ROC scores higher than 0.7.

Comment 15: (line 339) Typo: fonund -> found

Response: This typo was amended. The sentence now reads as follows (lines 351-352 of page 9 in the revised manuscript): “On average, 55 pathways are found to 

be described by a graphlet adjacency.”

Comment 16: (line 409) Typo: enables -> enable

Response: This typo is fixed. The sentence now reads as follows (lines 422-424 of page 11 in the revised manuscript): “Here, we illustrate how graphlet eigencentralities enable us to relate specific local wiring patterns of genes in a pathway with their individual biological function.”.

Comment 17: (line 431) explain what “MWU” stands for (Mann-Whitney U test). This is written in the Supplement but the main manuscript should be complete by itself.

Response: We agree with the reviewer and now formally define the abbreviation “MWU” in the main manuscript (lines 445-449 of page 12 in the revised manuscript): “To explain this, we perform a Mann-Whitney U (MWU) test comparing the distribution of the number of pathways that each cancer driver gene occurs in, with the distribution of the number of pathways that each non-cancer driver gene occurs in.”

Comment 18: Figure 5: color coding stated on the figure legend is not correct. All nodes appear red instead of red/yellow “nodes represent non-cancer-related genes (red) and known cancer driver genes RB1 and TP53 (yellow)”. Also, node ASF1A on the left figure should be moved, to not obscure the TP53-RB1 connection.

Response: Figure 5 has been rerendered so that nodes are correctly coloured (drivers/non-drivers), and node ASF1A does not obscure the edge between TP53 and RB1 any more.

Comment 19: Conclusion (lines 517-519): “Additionally, we show that cancer genes that can be uncovered by their pathway centrality are different depending on the graphlet eigencentrality”. Different as in what? Described mechanisms? Please, rephrase to clarify.

Response: We agree with the reviewer that this sentence was confusing and we amended it. The paragraph now reads as follows (lines 538-543 of page 14 in the revised manuscript): “Additionally, we show that by considering pathway centrality based on different graphlets, we can uncover complementary sets of cancer genes. We illustrate these results by a case study of the FSAHF pathway, where we show how graphlet adjacency, unlike regular adjacency, captures the central roles of cancer driver genes, RB1 and TP53. We conclude that graphlet eigencentralities allow us to capture different functional roles of genes in and between pathways.”

Comment 20: Conclusion (line 524): Delete “using networkss”.

Our response: This typo was removed from the manuscript. The complete sentence now reads (lines 546-549 of page 14 in the revised manuscript): “Further, the study of the centrality of nodes is not limited to biology, making our graphlet eigencentralities applicable in many disciplines that use networks as models, including physics, social sciences and economics.”

---

## [Decision Letter · Decision Letter 1]

9 Dec 2021

Graphlet eigencentralities capture novel central roles of genes in pathways

PONE-D-21-17376R1

Dear Dr. windels,

We’re pleased to inform you that your manuscript has been judged scientifically suitable for publication and will be formally accepted for publication once it meets all outstanding technical requirements.

Kind regards,

Ivan Kryven

Academic Editor

PLOS ONE

Additional Editor Comments (optional):

Reviewers' comments:

Reviewer's Responses to Questions

**Comments to the Author**

1. If the authors have adequately addressed your comments raised in a previous round of review and you feel that this manuscript is now acceptable for publication, you may indicate that here to bypass the “Comments to the Author” section, enter your conflict of interest statement in the “Confidential to Editor” section, and submit your "Accept" recommendation.

Reviewer #2: All comments have been addressed

2. Is the manuscript technically sound, and do the data support the conclusions?

Reviewer #2: Yes

3. Has the statistical analysis been performed appropriately and rigorously? 

Reviewer #2: Yes

4. Have the authors made all data underlying the findings in their manuscript fully available?

Reviewer #2: Yes

5. Is the manuscript presented in an intelligible fashion and written in standard English?

Reviewer #2: Yes

6. Review Comments to the Author

Reviewer #2: All comments have been addressed. The compiled revised PDF version still had a blurry version of Figure 1. Make sure the correct, sharpened version is uploaded to the system.

7. PLOS authors have the option to publish the peer review history of their article (what does this mean?). If published, this will include your full peer review and any attached files.

Reviewer #2: **Yes: **Evangelos Karatzas

---

## [Editor Report · Acceptance letter]

7 Jan 2022

PONE-D-21-17376R1 

Graphlet eigencentralities capture novel central roles of genes in pathways 

Dear Dr. windels:

I'm pleased to inform you that your manuscript has been deemed suitable for publication in PLOS ONE. Congratulations! Your manuscript is now with our production department. 

Kind regards, 

on behalf of

Dr. Ivan Kryven 

Academic Editor

PLOS ONE